# Selection Response Due to Different Combination of Antagonistic Milk, Beef, and Morphological Traits in the Alpine Grey Cattle Breed

**DOI:** 10.3390/ani11051340

**Published:** 2021-05-08

**Authors:** Enrico Mancin, Cristina Sartori, Nadia Guzzo, Beniamino Tuliozi, Roberto Mantovani

**Affiliations:** 1Department of Agronomy, Food, Natural Resources, Animals and Environment, University of Padua, Viale dell’Università, 16, 35020 Legnaro, PD, Italy; cristina.sartori@unipd.it (C.S.); beniamino.tuliozi@unipd.it (B.T.); roberto.mantovani@unipd.it (R.M.); 2Department of Comparative Biomedicine and Food Science, University of Padua, Viale dell’Università, 16, 35020 Legnaro, PD, Italy; nadia.guzzo@unipd.it

**Keywords:** milk, beef, morphology, breeding goals, local breeds

## Abstract

**Simple Summary:**

A selection index considering milk, beef, and functional traits is required by breeders of the local dual-purpose Alpine Grey breed because of the worsening of beef and other functional characteristics of the breed not yet accounted for in the present selection scheme. The present study has examined genetic correlations among different traits to investigate the possible breeding scenarios for this breed. Results indicate the need to shift selection toward a greater economic weight for beef and functional traits to improve the dual-purpose attitude and maintain the breed’s peculiar characteristics.

**Abstract:**

Selection in local dual-purpose breeds requires great carefulness because of the need to preserve peculiar traits and also guarantee the positive genetic progress for milk and beef production to maintain economic competitiveness. A specific breeding plan accounting for milk, beef, and functional traits is required by breeders of the Alpine Grey cattle (AG), a local dual-purpose breed of the Italian Alps. Hereditability and genetic correlations among all traits have been analyzed for this purpose. After that, different selection indexes were proposed to identify the most suitable for this breed. Firstly, a genetic parameters analysis was carried out with different datasets. The milk dataset contained 406,918 test day records of milk, protein, and fat yields and somatic cells (expressed as SCS). The beef dataset included performance test data conducted on 749 young bulls. Average daily gain, in vivo estimated carcass yields, and carcass conformation (SEUROP) were the phenotypes obtained from the performance tests. The morphological dataset included 21 linear type evaluations of 11,320 first party cows. Linear type traits were aggregated through factor analysis and three factors were retained, while head typicality (HT) and rear muscularity (RM) were analyzed as single traits. Heritability estimates (h^2^) for milk traits ranged from 0.125 to 0.219. Analysis of beef traits showed h^2^ greater than milk traits, ranging from 0.282 to 0.501. Type traits showed a medium value of h^2^ ranging from 0.238 to 0.374. Regarding genetic correlation, SCS and milk traits were strongly positively correlated. Milk traits had a negative genetic correlation with the factor accounting for udder conformations (−0.40) and with all performance test traits and RM. These latter traits showed also a negative genetic correlation with udder volume (−0.28). The HT and the factor accounting for rear legs traits were not correlated with milk traits, but negatively correlated with beef traits (−0.32 with RM). We argue that the consequence of these results is that the use of the current selection index, which is mainly focused on milk attitude, will lead to a deterioration of all other traits. In this study, we propose more appropriate selection indexes that account for genetic relationships among traits, including functional traits.

## 1. Introduction

The extension of milk and beef markets has contributed to the gradual decline of the appeal of local domestic breeds due to their low productive performance compared with specialized breeds [1]. Despite this, in recent years, local breeds have received an increased interest, as compared with cosmopolite breeds, they have better preserved functional characteristics (health, fertility, longevity, and rusticity). Local breeds are also often associated with the production of labeled foods (protected designation of origin/protected geographical indication), especially cheeses. [2]. Not considering only the economic aspects, these breeds have a link with the territory supporting rural/local economy and represent an effective resource of biodiversity [3]. In addition, local breeds are more adaptable to environmental changes than specialized breeds and they can also be bred in marginal areas and low-income environments [4]. Many local breeds have a dual-purpose attitude for milk and meat, but with differing emphases depending on traits’ local economic importance.

For these reasons, it is fundamental to ensure accurate breeding plans (aimed at improving the traits of interest) for local breeds, as previous studies have shown a negative genetic correlation between milk and meat traits [5,6].

The Alpine Grey is an autochthonous cattle breed of the central Alpine arc, widespread in Tyrol (Austria), South Tyrol (Italy), and neighboring Switzerland. Each country maintains its own herd book and independent breeding plans [7]. The Alpine Grey is generally well adapted to live and produce both milk and meat under challenging environments based on Alpine pastures. The present Italian Alpine Grey population accounts for 17,373 heads in 1737 farms (www.fao.org/dad-is/; update: 26 October 2020) distributed mainly in the provinces of Bolzano and Trento (85%). Milk production amounts to 5339 kg of milk per lactation with 3.75% of fat and 3.39% of protein, respectively. The average daily gain (ADG) of young bulls can reach 1.2 kg/day, and the carcass yields about 58% (ANAGA; www.grigioalpina.it, 20 April 2021). The breeding system for this Gray Alpine is generally constituted by small farms housing cattle during the winter months and releasing them to pasture in summer. The actual breeding goal to improve milk and meat traits is based on a selection index that assigns an economic weight of 24% to fat yield and 46% to protein yield. A further 20% of the economic weight is attributed to young bulls’ ADG, and the remaining 10% to rear muscularity (RM), which is evaluated as a type trait on primiparous cows. Therefore, the present breeding plan does not account for further functional and morphological traits that characterize the breed. Notwithstanding, these traits are required with increasing interest by breeders to maintain the typicality and rusticity of the breed.

Therefore, the aim of this study was to estimate the genetic correlations between milk, beef, and functional and morphological traits in the local Alpine Grey breed. This was done by analyzing the genetic response to selection under different scenarios in which different weights were applied to current and novel traits to be accounted for in the selection index. Particularly, traits investigated were milk, fat, protein, and somatic cell score. This latter trait was derived from the test-day (milk) dataset, whereas type traits were obtained from the scoring of primiparous cows and beef traits from young bulls at performance test.

## 2. Materials and Methods

### 2.1. Data Editing

All data were provided by the National Breeder Association of Alpine Grey cattle (ANAGA). The study used three different datasets, including milk, beef, and morphological traits. The milk dataset contained information on milk, fat, and protein yields (MY; FY, and PY, respectively; kg/d) and somatic cell counts (no./mL), with an average interval of 4 weeks between test day collection. Dataset of morphological evaluations contained 21 type traits routinely scored on primiparous cows when aged about three years (36.9 ± 5.0 months). The dataset on beef attitude was obtained from performance test data and contained the average daily gain (ADG), an in vivo estimate of carcass yield (CY), and muscularity traits (SEUROP scale) carried out by skilled classifiers on young bulls aged about 12 months.

The milk dataset initially contained 1,134,032 individual test-day (TD) productions routinely collected from 1997 to 2018 following the Italian official milk recording system. The number of somatic cells/mL was converted into the normally distributed somatic cells score (SCS) according to [8]. As first data editing, the TD records with missing values, and the ones recorded when days in milk (DIM) was under 5 d or over 305 d from calving were removed. In additions, only TD belonging to lactation from 1 to 3 were retained. Values for MY, FY, and PY outside the mean ± four standard deviations within parity and lactation phase (considering 15 d intervals) were taken away from the data set as outliers. Among the remaining records, only those belonging to cows with age at calving between 21 and 44 months at first parity, between 32 and 60 months at second parity, and between 44 and 76 months at third parity were retained for analysis. Furthermore, only lactations with a first TD carried out within 45 days from calving and including at least four records were kept for further analysis: the reason for this was that functional controls of the cows are limited during the first 45 days, with the lactation peak occurring later in this breed [5]

Lastly, only records belonging to herd-TD with at least two observations could enter the final dataset. At the end of the editing process, a final dataset with 406,918 TD records belonging to 58,041 lactations and 29,219 cows was used. The pedigree file contained 49,389 animals, tracing back up to the sixth generation (complete generations).

Morphological traits are routinely measured once in the lifetime, around the time of first calving. Data initially consisted of 14,669 observations of 21 type traits scored on a scale of 1 to 50 points by trained classifiers during 2010–2018. The edited dataset contained 11,318 final records belonging to the same number of cows and 32,494 animals in the pedigree file. Records allowed to the final dataset were scored between 5 to 305 days in milk (DIM) and considered cows with age at first calving between 21 and 45 months. An exploratory factor analysis was carried out on all 21 type traits applying the varimax rotation [5,9]. Varimax rotation allows for a better interpretation of the biological meaning of each factor. The process consists in adjusting (rotating) the coordinates obtained from principal components analysis (PCA). This adjustment is based on maximizing the variance shared among components, increasing the squared correlation of items related to one factor, while decreasing the correlation to any other factor.

Then, further analysis retained three main factors describing: udder volume (UV; Factor 2, that is F2 in Figure 1), udder conformation (UC; Factor 3, that is F3 in Figure 1), and rear legs (RL; Factor 7, that is F7 in Figure 1), as these three were the main factors of interest to breeders in terms of the genetic index, as they are latent factors connected with the production of milk (F2), with the health of the udder (F3), or with the aptitude for grazing (F7).

These factors showed eigenvalues greater than one. They were named based on the biological meaning of the linear type traits showing the loading coefficients highest than an absolute threshold of |0.45|, a way of proceeding also applied in other previous studies [5,6].

The milk trait dataset and the morphological dataset were combined to perform correlation analysis. The two datasets had in common 9145 animals representing about 30% of the animals in the milk dataset.

The performance test dataset contained 749 records collected from 1988 to 2018 and belonged to the same number of bulls grouped by age and accounted for 6266 animals in the pedigree file. The final dataset included only contemporary groups of young bulls consisting of at least three animals. The average daily gain (ADG) was obtained as a linear regression of monthly weight on age. Further analysis considered only regressions with a coefficient of determination of at least 0.95. The number of weight-age couples used for the linear regression was 12 for each bull, and the average age at the beginning and the end of the performance test were respectively 50 ± 12 d and 356 ± 11 d. The in vivo visual appreciation of fleshiness, evaluated using the SEUROP scale, and the in vivo carcass yields scored independently by two evaluators, were obtained at about 375 ± 16 d of age. For each trait, the analysis considered the average of the two evaluations. The in vivo SEUROP score considering the grades S, E, U, R, O, and P, from the best to worst conformation, was further subdivided into + or − subclasses as in [10]. The scores were then transformed in a linear scale from 80 (corresponding to a grade of P) to 130 (corresponding to S), adding or subtracting 3.33 points to the full class when necessary; that is, for an R+ grade, the score was 103.33, whereas for the U− it was 106.67. The whole numeric interval, ranging from 76.67 to 133.33, was considered as continuous. The carcass yield was expressed as a percentage and was an in vivo appraisal of the predicted carcass incidence at slaughter.

### 2.2. Models

Milk traits were analyzed using the following test-day model:$$y_{\mathrm{ijklmno}}{=\mathrm{HTD}}_{i}{+\mathrm{LN}}_{j\;}{+\mathrm{GL}}_{k}+\overset{3}{\underset{r=1}{\sum }}\varphi_{r}{\times \;\mathrm{AP-LN}}_{l}+\overset{3}{\underset{r=1}{\sum }}\psi_{r}{\times \;\mathrm{MP-LN}}_{m}{+\mathrm{Pe}}_{n}{+a}_{n}{+e}_{\mathrm{ijklmno}};$$
where y_ijklmno_ is the individual test-day o^th^ record (milk, fat, protein, and SCS) of the nth cow; HTD_i_ is the fixed effect of the herd-test-day (90,012 levels); LN_j_ represents the fixed effect of lactation number (3 levels, corresponding to the first three lactations); GL_k_ is the fixed effect of kth gestation length class (18 classes with 1 meaning no gestation and further classes accounting for 15-d intervals, from 1 to 240 d of gestation); AP-LN is the fixed effect of lth age at parity within lactation (42 classes in total); MP-LN is the fixed effect of the m^th^ month of parity (36 classes corresponding to single months of a year within each j lactation); Pe is the random permanent environmental component, N (0,σ^2^_pe_); a is the additive genetic component, N (0,σ^2^_a_); and e_ijklmno_ is the random residual term, N (0,σ^2^_e_). Fourth-order Legendre polynomials described the shape of the lactation curve for the fixed effects of AP-LN and MP-LN, with φ and ψ as fixed regression coefficients for the Legendre polynomial of order r varying between 0 and 3.

Factor analysis for all the morphological traits was carried out to reduce the number of traits and avoid redundant morphological measurements (see also Data Editing section). The traits included in the analysis are described in Table 1. The factor analysis was performed using the “psych” package of R [11]. The varimax rotation method was used [12]. Latent variables with eigenvalues ≥ 1 were retained for further analysis. The factor score originated from every latent variable was considered as a new trait. According to traits of main interest expressed by the national breeders’ association, only three of seven latent factors were considered for subsequent analyses. They were called Factor 2 (F2), corresponding to udder volume; Factor 3 (F3), that is udder correctness, and Factor 7 (F7), rear legs (Figure 1). Together with these factors, the subsequent analysis also included the linear scores for rear muscularity (RM) and head typicality (HT), respectively considered as a beef trait and a functional trait for the breed.

The five morphological traits, two linear and three factor scores, were analyzed with the following model:$$y_{\mathrm{ijklm}\;}{=\mathrm{HY}}_{i}{+C}_{j}{+\mathrm{AC}}_{k}{+\mathrm{DIM}}_{l\;}{+a}_{m}{+e}_{\mathrm{ijklm}};$$
where y_ijkl_ is one of the five morphological traits; HY_i_, C_j_, AC_j_, and DIM_k_ are respectively the fixed effects of the herd-year (i = 3 318 levels); the classifier (j = 67 levels); the age at calving (k = 12 classes: <21 months, from 21 to 45 using 2-months intervals); and the days in milk (l = 20 classes from 5 to 305 days after calving and using 15-days intervals a_m_ is genetic random additive effect of animals N (0, ^2^_a_); and e_ijklm_ is the random residual term, N (0, ^2^_e_).

Regarding the beef traits, the following animal model was implemented:$$y_{\mathrm{ij}}{=\mathrm{GP}}_{i}+a_{j}+e_{\mathrm{ij}},$$
where y_ij_ is a performance test phenotype for ADG, SEUROP, or CY; GP represents the categorical fixed effect of the contemporary group (i = 142 levels); a_j_ is the random additive genetic effect of the young bull j; and e_ij_ is the random residual term.

### 2.3. Variance Component Estimates and Model Assumptions

To estimate the (co)variance components, a Gibbs sampling algorithm was used, and the analysis was performed with the *gibbs3f90* program [13].

The program generated a total number of 480,000 samples and considered an initial burn-in of 30,000; one of every 150 chains was retained. A Gaussian distribution for all effects was considered.

Flat priors were used for all fixed effects, and null means and normal distributed priors were used for permanent environment, additive genetic, and residual terms, with this matrix notations:a∼N(0,G⊗A); pe∼N(0,Pe⊗I);e∼N(0,R⊗I);
where A represents the relationship matrix obtained from pedigree, and I is an identity matrix.

Heritability was obtained from variance components estimated by applying single-trait models, while genetic and phenotypic correlations from bi-traits models were obtained by merging the three different datasets in pairs. The covariance matrices used in the bi-traits analysis were as follows:$$G=\left|\begin{array}{cc} \sigma_{a1}^{2} & \sigma_{a1a2}\\ \sigma_{a1a2} & \sigma_{a2}^{2} \end{array}\right|;\;\mathrm{Pe}=\left|\begin{array}{cc} 0\left[\sigma_{\mathrm{pe}1}^{2}\right] & 0\left[\sigma_{\mathrm{pe}1\mathrm{pe}2}\right]\\ 0\left[\sigma_{\mathrm{pe}1\mathrm{pe}2}\right] & 0\left[\sigma_{\mathrm{pe}2}^{2}\right] \end{array}\right|;\;R\left|\begin{array}{cc} \sigma_{e1}^{2} & 0\left[\sigma_{e1e2}\right]\\ 0\left[\sigma_{e1e2}\right] & \sigma_{e2}^{2} \end{array}\right|;$$
where G is the matrix of additive genetic (co)variances σ^2^_a1_, σ^2^_a2_, σ_a1a2_ of traits 1 and 2, Pe is the matrix of permanent environmental (co)variances σ^2^_pe1_, σ_pe1pe2_, σ^2^_pe2_, and R the matrix of residual (co)variances σ^2^_e1_, σ^2^_e2_ and σ_e1e2_ of traits 1 and 2. When different datasets were merged, residual (co)variance was set to zero because the traits were recorded in different moments. In single traits analysis, Pe was not considered (i.e., beef and morphological datasets) because obtained only once in life. Nevertheless, when morphological or beef traits were analyzed with milk traits, a covariance σ_pe1pe2_ was included to provide a better estimate of the permanent environment component for milk yields traits, according to [14]. From a biological point of view, this $\sigma_{\mathrm{pe}1\mathrm{pe}2}$ represents the relationship between traits due to the common environment represented by each individual.

### 2.4. Estimated Selection Response

A final step consisted in calculating the theoretical multivariate response to selection (R) under different weights for each trait considered as a breeding goal. The response to selection (R) is the change of the phenotypic mean during a generation for a specific or a group of selected traits. The theoretical multivariate response to selection [15] was calculated according to [16] using the following formula:$$R=\left(i/\sigma i\right)\cdot b^{\prime }\cdot P^{-1}$$
where *i* is the selection intensity set to 1.755 as in [6] corresponding to a proportion of 0.10 selected animals in the whole population, assuming a normal distribution; *σ_i_* is the SD of the selection index, obtained as $\sigma_{i}={\left(b^{\prime }\;P\;b\right)}^{1/2}$; *b* is the vector of the weights for selection index and b’ its transpose, with $b=P^{-1}\;Ga_{s}$. In this formula, P and G are the phenotypic and the genetic (co)variance matrices, respectively, and a_s_ is the vector including the economic weights of traits. In this vector, P and G have the same meaning as in in the formula above, and as is the vector including the standardized economic weights of traits. As in [16], the relative emphasis of the traits in the selection index was intended as a proportion a of the trait’s standardized economic value (i.e., as = a × σ_a_) compared with the sum of all standardized values of all the traits accounted in the index. A final standardized response to selection (R_dsi_) was calculated as $R_{dsi}=R\;/\;\sigma_{i}.$


Eight different scenarios (Si) to estimate the selection response were simulated. The first scenario (S1) considered the current selection emphasis given to traits under routinely selection practices: 70% for milk traits (24% to fat yield and a 46% to protein yield) and the remaining 30% attributed to ADG (20%) and RM (10%) beef traits. All the other traits (that are all the traits mentioned above of the three datasets) had a selection emphasis of zero since they are not included in the selection index. They are indirectly selected due to the genetic correlations they have with the traits under direct selection. Scenario 2 and 3 (S2 and S3) had the same selection weights of S1, but in S2, the genetic gain for SCS was restricted to zero according to previous studies [6,16]. This restriction was done to prevent an increase in SCS since it could imply a detriment in udder health conditions. Similarly, in S3, the genetic gain for both SCS and RM were restricted to zero. The RM variation was restricted to zero to prevent a worsening of the traits due to the negative genetic correlations occurring with milk yield traits, as reported below. In this latter case, 30% of weight attributed to beef traits was entirely shifted to ADG. The S4 and S5 provided less emphasis on milk attitude (65% for both scenarios), and the remaining 35% was divided in different manners. The S4 assigned 15% of the weight to RM, 5% to SEUROP, and 5% to CY, meaning a total of 25% of the weight on beef traits, and attributed 7% of the weight to F3-UC and 3% to HT. In S5, less credit to morphological traits was given with respect to S4, i.e., 3.5% of the weight to F3-UC and 1.5% to HT, with a corresponding increase of RM to 20% of the weight. In S6 and S7, the milk traits’ weight was further reduced to 55%. The morphological traits F3-UC and HT received the same weight as in S4 (for S6), and S5 (for S7), while the beef traits SEUROP and CY received a 10% of weight each in both S6 and S7. Last, in S8, milk traits were set to 70%, and beef traits to 30%, specifically 20% to RM and 5% each to SEUROP and CY, but restriction to zero were imposed for SCS and morphological traits (F3-UC, F7-RL, and HT) to prevent a detriment in their genetic variation.

## 3. Results

### 3.1. Descriptive Statistics and Factor Analysis

Mean, standard deviation, maximum and minimum values for all the studied traits are shown in Table 1. Milk, fat, and protein TD yields indicated a daily production of 16.3, 0.62, and 0.56 kg/d, respectively. Regarding SCS, a mean of 2.33 was found, i.e., approximately 62,760 cells/mL, suggesting an excellent value for mammary health in Grey Alpine. Almost all morphological traits present an average value close to 28 points, except the teats position—rear view (23.5 points). Young proved bulls presented a mean ADG of 1.15 kg/d, with a carcass yield of 56% and an average SEUROP conformation score of 103 points, corresponding to an R+ score.

Figure 1 reports the factor analysis’s main results that indicate a quite clear biological interpretation of latent factors based on loading coefficients with an absolute value greater than 0.45. Indeed, muscularity, udder volume (UV), udder conformation (UC), general aspect, feet correctness, teats, and rear legs (RL) have been identified after varimax rotation pattern as the biological meaning of factors F1, F2, F3, F4, F5, F6, and F7, respectively (Figure 1). These latent factors express an amount of variance of 2.68, 2.38, 2.03, 1.91, 1.62, 1.25, and 1.20, respectively (Table 1). The F1 included the following traits (minimum loading coefficient of |0.45|: strength/robustness with a loading coefficient of 0.86, rear muscularity (0.90), and fore muscularity (0.91). The F2 included some morphological traits regarding udder volume, such as: fore udder length with a loading coefficient of 0.67, and rear udder attaches (height; 0.83; and width; 0.86). The F3 was related to udder conformation and accounted for: fore udder strength (0.64), suspensory ligament (0.66), udder depth (0.81), and udder symmetry (0.47). The F4 contained traits describing the general aspect of the individual: thinness (0.63), shoulders (0.73), dorsal line (0.69), and head typicality (0.58). The F5, accounting for the leg correctness, comprised pastern and foot angle (both with a loading coefficient of about 0.85). The F6, teats, included traits related to teats evaluation, i.e., teats length (0.76), both side (0.52) and rear view (−0.54) of teats position, although with opposite sign. Last, F7 accounted for rear legs viewed by side (0.77) and back view (−0.48). F2, F3, and F7 was retained for further genetic analysis. Overall, these three factors accounted for 27.3% of the total variance of traits (Table 2) and allowed a clear interpretation of the latent variables. Each of them was further analyzed as a factor score, which resumes the information of the traits included using a standardized phenotypic variable [5]. In addition to the three factor scores, single linear type traits of rear muscularity (RM) and head typicality (HT) were analyzed.

In fact, the first one represents a trait presently under selection, and the second one is a trait in which breeders shows a strong interest because it is part of the breed’s typicality: a qualitative assessment of the cranial shape according to the Gray Alpine Herd Book (http://www.grigioalpina.it/wp-content/uploads/2015/10/Norme-tecniche.pdf, 6 April 2021).

### 3.2. Genetic Parameters and Genetic Correlations

Table 3 reports the variance components and genetic parameters estimated in single-trait models. Note that not all traits possess all three components, i.e., genetic, environmental, and residual: for example, udder volume factor, since it is tested once, does not possess σ^2^_pe_ component. Compared to morphological and beef traits, milk traits showed generally lower heritability values, ranging from 0.218 (milk yield) to 0.133 (SCS). Morphological traits had medium-high heritability, i.e., near to 0.30, with head typicality that presents the highest h^2^ in this group of traits (0.374), while F3-UC (udder conformation) showed the lowest value of heritability (0.238). Beef traits measured early in life on young bulls at performance testing station showed the highest h^2^. SEUROP and CY (carcass yield) reached a value of 0.376 and 0.501, respectively. Conversely, AGD presented a medium heritability value of 0.282.

Table 4 showed the genetic and phenotypic correlations between each trait pair considered in the study (full table with HPD is available in Appendix A). High genetic correlations (>0.75) were observed within milk traits, except SCS, which was mildly correlated from the genetic point of view with milk yield traits. High genetic correlations were also found between beef traits; mainly, SEUROP and CY presented a genetic correlation greater than 0.90. Regarding the genetic correlations within morphological traits, low values were generally observed, except for some negative correlations between F2-UV and F3-UC (−0.208) and between rear muscularity and F2−UV (−0.319). On the other hand, a medium but positive genetic correlation was obtained between rear muscularity and F3−UC (0.346). Considering the genetic correlations between different groups of traits, F2−UV had positive correlations with all milk yield traits, whereas F3−UC had a negative association with milk yield. F2−UV also had negative correlations with beef traits SEUROP and CY. Another trait related to the beef attitude is the morphological trait rear muscularity of primiparous cows (RM). This trait had a medium but negative genetic relationship with milk yield traits (from −0.158 to −0.458). Interestingly, despite ADG carcass yields, and SEUROP were strongly correlated with RM, they were not so negatively correlated as RM with milk yield traits.

Phenotypic correlations follow the same trends as the genetic ones, but their absolute values were slightly lower, especially among different groups’ traits.

### 3.3. Genetic Trends and Response to Different Selection Scenarios

Table 5 reports the economic weights assigned to each scenario and Table 6 the weight obtained after restriction to zero. In the current selection scheme (S1), great standardized selection responses were obtained for all milk traits (including F2−UV; Figure 2). The protein yield (PY), as expected, had the maximum value (0.47), corresponding to a genetic progress of 0.067 kg of protein per generation (data not shown). All other traits (morphological and beef) showed a negative genetic trend (Figure 2; S1). F3−UC presented the worst selection response (−0.32). ADG and SEUROP resulted in the only beef traits with non-negative selection response (0.29 and 0.05, respectively).

When restrictions were applied to SCS (S2), corresponding to null genetic progress for this trait, the expected genetic gain for milk traits slightly declined: for PY from 0.47 to 0.40 and from 0.32 to 0.29 for FY. About morphological traits, F3−UC showed a further decrease from −0.32 to −0.36. Similarly, RM resulted in less negative genetic variation (from −0.16 to −0.04 standardized units) than in S1, almost null genetic progress. In the scenario in which restriction was applied on RM (S3), a similar situation than in S2 was seen, although with a small reduction of standardized genetic gain for milk and a small increase for all beef traits were observed, i.e., a non−negative variation for RM, positive increase for ADG, SEUROP, and CY (Figure 2, S3). On the other hand, the expected genetic progress for all morphological traits remained almost unchanged with respect to S1 and S2. Indeed, F2−UV still showed a small increase, F3−UC remained negative, such as HT and F7−RL, although with a lower magnitude than for F3−UC. In S4 and S5, despite the small reduction of fat and protein yields and beef traits in favor of morphological traits, a small but favorable increase of FY and PT was still observed, because of the lower incidence of beef traits, but SCS increased negatively. However, in both scenarios, F3−UC and HT genetic gain resulted less negative than in the previous ones (Figure 2), while F7−RL showed a small negative increase. In S5, notably, RM resulted non-negative. In S6 and S7, there was a further reduction of milk traits favoring beef traits but maintaining the same weights for morphological traits as in S4 and S5. Thus, both FY and PY were reduced, and beef traits increased compared to the previous scenarios. S6 and S7 showed the best genetics progress for beef traits (>0.20, >0.30, >0.45, and >0.40 for RM, ADG, SEUROP and CY, SEUROP, respectively; Figure 2). These were the first scenarios in which RM had a positive selection response. On the contrary, milk traits presented the worse genetic gain as compared to other scenarios. In both S6 and S7, the SCS increased negatively, but slowly than in S4 and S5. Response to selection of F3−UC was less negative than in the previous scenarios, while F7−RL resulted more negatively affected (Figure 2). Last, in S8, milk yield genetic gain was maintained at the same level as in S6 and S7, but morphological traits resulted in a non-negative variation, and beef traits improved, particularly RM in primiparous cows. Additionally, SEUROP and CY of young performance-tested bulls showed a positive increase (Figure 2). In general, F2−UV, not weighted in any scenario, followed the same trend as milk yield, i.e., increasing when milk yield increased, showing a selection response over to 0.20 standardized units. Notwithstanding, reducing milk traits’ economic weight contributed to an adverse selection response for F2−UV in scenarios S6, S7, and S8.

## 4. Discussion

### 4.1. Heritability

The Grey Alpine represents a perfect example of a local breed with a dual-purpose attitude, as it shows good productive performances for both milk and beef traits. In our study, heritability of milk yield was lower than other traits analyzed, but this is because milk yield was analyzed as test-day records, which are recognized to give lower heritability, because of the high environmental variance. Moreover, it is also due to the nature of the data, i.e., longitudinal observations instead of data recorded once in life. Fat yields showed lower heritability than protein yields, although both PY and FY had about the same additive genetic variance. However, FY’s residual variance was almost double compared to that of PY, reducing the heritability. Many studies have reported that fat is much more affected than protein by external factors, such as the feeding regimen [17,18], and that is in agreement with the greater residual variance for FY than for PY. On the other hand, the Grey Alpine showed heritability valued for milk traits similar to those reported for other dual propose/local breeds, as Italian Simmental, Rendena, and Valdostana. The Italian Simmental presented a heritability value of 0.18, 0.13, and 0.17 for milk, fat, and protein yields, respectively [19]. In the Rendena local breed, heritability levels for these traits were 0.188, 0.157, and 0.165 [6], and similar values were also found in the Valdostana breed, for which heritability estimates were 0.198, 0.132, and 0.169 [20]. In general, heritability for milk traits resulted slightly greater than in specialized breeds, like Holstein (e.g., 0.108 for MY in Italian Holstein; [21]). This could be related to the fact that in dual-purpose cattle, milk traits have been subjected to less selective pressures over time than dairy cattle [3].

Although many studies considered SCS as a low heritability trait (h^2^ = 0.08 on average), especially in Holstein cows [22], a slightly greater value of 0.133 was found in this study. Still, the lower selection pressure could be identified as the possible cause of such greater than expected estimates. However, for other local cattle breeds of the Alpine area, Rendena, and Valdostana (Aosta Chestnut), heritability estimates were closer to those observed in cosmopolitan breeds, i.e., 0.08 [20].

Factor analysis allowed to characterize any factor with an explicit biological meaning due to the orthogonalization of loading coefficients, performed by varimax rotation, that maximizes factor independence [12]. According to [23], factor loadings are one of the best approaches for selection when a lot of different traits can be easily combined because of their collinearity. Factor loadings indeed allow summarizing the information from a multi-trait analysis by concentrating the traits into single information, avoiding the use of highly correlated measures. In this study, F2 was entirely explained by all the udder attaches (length of the fore udder and length and width of the rear udder); a greater value of the factor loading indicates a wider dimension of udder attach, directly linking the factor to the volume of the udder. The F3 included the other udder traits connected to udder “health”, like the strength of the suspensory ligament, the udder depth, and the udder symmetry. These three traits describe the mammary apparatus’s conformation and assume an increasing value of F3 with an increased score of the three traits. Last, F7 describes the posterior rear legs conditions, including the side and the back view of rear legs. These two traits, characterized by intermediate optimum values, entered the factor with opposite sign (i.e., positive the rear legs side view, and negative the rear legs back view), because the sickle-hocked defect, associated with a greater score, is often associated to the cow-hocked defect, assuming the lowest scores in Alpine Grey morphological evaluation (www.grigioalpina.it/?lang=en, Date of access: 30 March 2021).

Overall, morphological traits showed a medium-high value of h^2^, as widely reported in the literature [24,25,26], including dual-purpose cattle [5]. Recent studies [27,28,29] demonstrated that the higher heritability values could be due to the greater number of gene clusters involved in biological processes relevant for udder morphology.

A wide range of studies have been carried out for morphological traits in specialized breeds, and generally, the heritability resulted lower than those estimated in this research. Regarding morphological traits evaluated in dairy cattle (udder volume, udder conformation, also the leg), heritability values of 0.14, 0.08, and 0.07 were found in Holstein; also, a 0.18 for udder volume was reported [30,31] and as a mean of 0.22 for other traits regarding udder conformation [30,31]. For beef conformation traits, a heritability of 0.40 has been reported in both Brown Swiss and Red and White breeds [32], but in Italian dual purpose breeds, values similar to those of the present study were found. For muscularity, udder volume and udder conformation heritabilities of 0.314, 0.166, and 0.169 were found in Valdostana [5], whereas Rendena showed higher heritability values of 0.359, 0.260, 0.267, possibly due to the different nature of the factorial score in these breeds as compared to the Alpine Grey. On the other hand, in beef cattle, similar heritability estimates have been reported for head typicality [25,26].

The high heritability estimates observed in this study for performance test traits compared to the morphological and milk traits were commonly observed even in other dual-purpose or beef breeds, like the Piedmontese (e.g., heritability of 0.47 for ADG) [6,32,33,34,35].

### 4.2. Genetic Correlations

As expected, milk yield traits showed strong phenotypic and genetic correlations among them, as both protein and fat productions depend on the amount of milk produced. Somatic cell score showed a low-positive genetic correlation with milk traits, confirming previous findings [36], where the independence of traits was demonstrated by genomic analysis indicating the presence of different genes and loci under the traits [37,38]. Udder volume (F2) showed a positive correlation with all milk traits, including SCS (a positive correlation was also reported in other studies [39]). On the contrary, F3 (udder conformation) presented a negative correlation with F2, and consequently, with all other milk yield traits. The genetic improvement for milk production leads to an indirect increase of udder volume that causes damage to its conformation. Similarly, [6,30,31] found a negative genetic correlation of about −0.3 between udder conformation and udder volume for Italian Brown Swiss, Rendena, and Valdostana cattle. An impressive result was discovered by analyzing the genetic correlation of SCS with F2−UV and F3−UC, which resulted positive, suggesting a detriment in udder health for increasing udder volume and conformation values. A similar result was also found in the Rendena breed for udder volume [5], but it was the opposite for udder conformation. Rear legs (F7) and head typicality had genetic correlations not different from zero, either with milk or beef traits, considering that in all cases, the HPD95% included zero. The only positive correlation was observed between SCS and F7−RL, meaning that an increase in inflammatory udder status is associated with an impairment of rear legs and possibly a general unhealthy animal status [40]. Milk traits and F2−UV had a negative correlation with rear muscularity, whereas the positive medium correlation between F3−UC and RM can be explained considering the negative correlations that both these traits showed milk yield because they both resemble a typical aspect observable more frequently in muscular cows. Positive correlations between muscularity and udder correctness have been previously reported [5]. Muscularity score in cows showed a strong positive genetic correlation with all performance test traits, which agrees with other studies [6,10], despite a negative genetic correlation sometimes found between muscularity and ADG [21,33] The negative correlations between milk and beef traits have been identified in the different asset of genes involved in metabolism regulation, catabolism of collagen, and myogenesis compared to milk synthesis [41]. Other studies have reported this negative genetic correlation, estimating a similar correlation coefficient to our study, e.g., in Brown Swiss and Swiss Simmental cattle [39] or Italian Simmental [40]. On the other hand, slightly lower negative genetic correlations were found by Croué et al. [42], comparing the postmortem SEUROP with milk, fat, and protein yields in French dual-purpose cattle breeds (Montbeliarde, Normande, and Simmental).

### 4.3. Genetic Response under Different Selection Scenarios

Multivariate response to selection was calculated to properly account for different traits in the aggregate selection index of the breed. This index is made by assigning a different economic weight to EBVs of each target trait [43].

A proper knowledge of the true genetic relationships among target traits, and of the expected response under different selection pressures can help properly drive selection decisions and therefore the genetic trend of traits.

The present selection scheme (S1) produces the greatest growth of milk traits in terms of standardized genetic progress due to the high weight accounted by fat and protein yields in the selection index (70% of total). The positive selection response for milk yield is due to the favorable and strong genetic correlations with the FY and PY. The present scheme produces a negative increase of the SCS, negatively affecting udder conformation, the muscularity measured in primiparous cows, and head typicality. S1 produces also a positive increase in udder volume, ADG measured on bulls, and to a less extent, SEUROP scores on young bulls. A steady-state is detectable for rear legs and estimated CY on performance-tested young bulls.

The considerable selection response attainable for F2−UV, despite not being directly accounted for in the selection index in S1, is due to the strong genetic correlation of this composite trait with milk yield traits, since a large udder volume allows an increased milk production. In the current selection index, all the beef traits except ADG showed a negative (RM) to almost null (SEUROP and CY) selection response, because the economic weight for beef attitude was attributed only to ADG (i.e., 30%). Overall, the present selection index does not reflect the goal of selection for the dual-purpose attitude in the Alpine Grey cattle breed. A different situation was observed in the current selection response of another Italian dual-purpose cattle, the Rendena, in which an economic weight for beef attitude is due to all the traits accounted for in this study [5]. Nevertheless, negative response for RM is also produced in this breed due to the strong antagonistic correlations with milk traits. The positive response for SCS, observed in this study as also in [5] is undesirable for selection since it means a detriment in udder health.

The restriction for maintaining unchanged SCS in S2 produces a slightly negative effect on milk yields and slightly increases the response for the beef traits measured in performance-tested bulls. Despite this, the negative genetic correlations between muscularity and milk traits still led to an adverse effect of RM selection, limiting a proficient selection for the breed’s dual-purpose attitude. Despite the neutral selection response for SCS, udder conformation (F3−UC) worsens with respect to S1, highlighting the need for further investigations on these traits that are both indirectly linked to udder health.

When a restriction toward unchanged SCS and muscularity was analyzed (S3), a decline in milk yield progress was observed, underscoring the importance to reduce milk yield growth despite a selection goal more oriented toward the dual-purpose attitude. Although CY and SEUROP in young bulls were not directly selected, an increase in the standardized genetic response for these traits is detectable because of the positive correlation with muscularity.

In the fourth scenario (S4), an increase in milk yield response is observed, despite a 5% reduction of its weight, but beef trait response was like S2 and S3. However, SCS and morphology generally showed a negative response to selection, slightly more negative in the fifth scenario (S5), where they received a lesser economic weight, in favor of beef traits. Such traits on the other hand increased, particularly muscularity in primiparous cows which was slightly positive for the first time. In subsequent scenarios, milk yield was negatively affected due to the reduced economic weight in the selection index (as in S6 and S7) or the introduction of constraints in non-negative growth of traits negatively correlated to milk, fat, and protein yields. In the last three scenarios, muscularity was greatly increased as compared to the previous scenarios. S6 and S7 were the scenarios in which beef had more emphasis in the selection process. On the other hand, S8, notwithstanding the 70% of the weight on milk traits, produced results like S6 and S7, due to the greater incidence of beef and morphology. These latter scenarios should be considered technically more balanced toward the dual-purpose attitude, although they could be considered not as economically convenient due to the high commercial value of milk as compared to other traits.

In all scenarios proposed, the udder volume (F2) was always not directly selected due to its high correlation with fat and protein, and milk yield. Nevertheless, this trait showed a negative selection response in the last three scenarios due to the negative correlations with beef traits. Regarding other morphological traits (F7−RL and HM), following breeders’ suggestions, it could be important to ensure a non-negative or slightly negative genetic progress for these traits, as observed in most cases. Regarding performance test traits, high economic weight for ADG, as in S1, is meaningless. ADG has a less economic interest than CY and SEUROP (ANAGA, personal communication) that are also less negatively correlated with milk traits. If non-negative genetic progress for SCS, but especially for F3−UC and RM, is considered a priority, a necessary reduction of milk traits’ genetic progress occurs.

## 5. Conclusions

In conclusion, due to the complex structure of genetic correlations among traits and the large number of negative genetic correlations listed above, selection index including all the important aspects for the breed (milk, beef, and morphology) can be considered the best compromise. As expected, milk and beef traits have negative genetic correlation, and the situation becomes more complicated if morphological traits are included in the selection index.

However, the present selection index (S1) produces a detriment of beef attitude in the medium-long term and a loss of some peculiar and important functional characteristics in the breed. For these reasons, a selection index more oriented toward the beef attitude, without worsening some morphological characteristics appreciated by breeders, can be considered more appropriate, despite the reduction of the expected response for milk yield.

The best scenario cannot be uniquely identified, because the economic values of a standard deviation of different traits are not equally predictable. This is particularly true for morphological traits for which intrinsic economic value is often hard to measure (i.e., F7), or it has a null value, but it is of great importance for maintaining the typicality of the breeds (i.e., HT).

In this regard, the authors however suggest scenario 7 as the most suitable for selection in the Alpine Grey breed. In fact, scenario 7 allows a slight genetic progress for both productive traits (i.e., milk and meat), while preserving the dual attitude of the breed. Furthermore, this scenario guarantees the maintenance of the functional characteristics of this breed.

## Figures and Tables

**Figure 1 animals-11-01340-f001:**
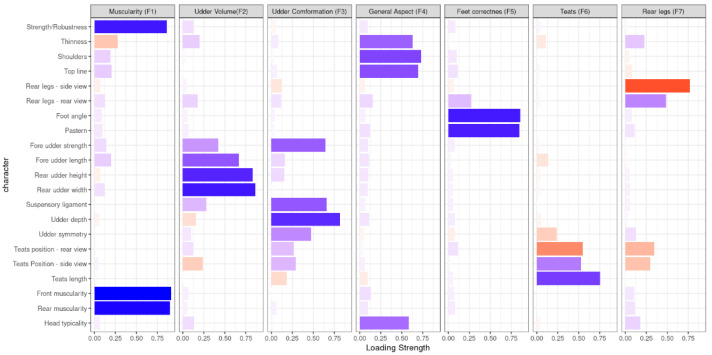
Loading coefficient (LC) of individual morphological traits within the seven latent factors extracted from the factor analysis (i.e., with eigenvalue > 1) after the varimax rotation. Only LC ≤ −0.45 or ≥0.45 have been reported. Blue bars represent positive loading coefficients, red bars negative loading coefficients.

**Figure 2 animals-11-01340-f002:**
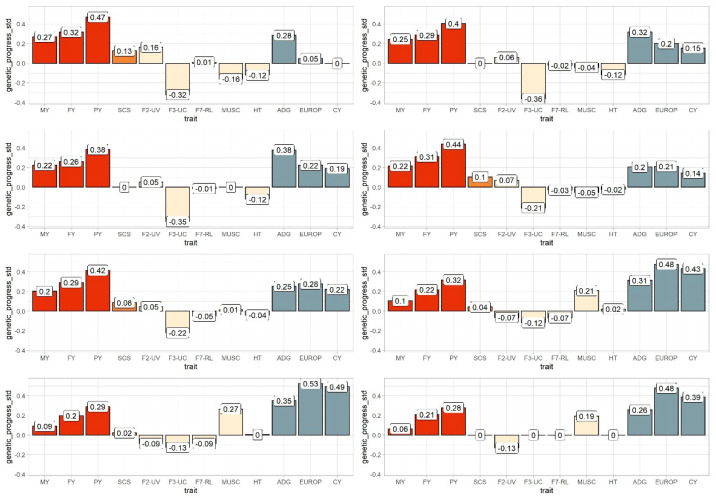
Standardized genetics progress (y axes) for 12 traits studied considering 8 different scenarios (from S1 to S8) attributing different weights to specific traits in the possible selection index. Red bars represent milk, fat, and protein, orange bars SCS, light-yellow morphological traits (both factors and single trait analyzed), and grey bars beef traits (type rear muscularity or performance test traits). Traits’ abbreviations are reported in Table 1.

**Table 1 animals-11-01340-t001:** Descriptive statistics of analyzed traits.

Traits	Mean	SD	Minimum	Maximum
Milk traits				
Milk yield (kg/d)	16.30	5.36	0.60	45.20
Fat yield (kg/d)	0.62	0.21	0.02	2.163
Protein yield (kg/d)	0.56	0.17	0.02	1.49
Somatic cell score (points)	2.33	1.86	−3.64	10.84
Linear Type traits (points; scale 1–50)				
Strength/Robustness	29.15	6.67	Tight and weak	Large and strong
Thinness	26.42	5.46	Heavy and coarse	Thin and sharp
Shoulders	28.80	5.65	Loose	Smooth and adherent
Top line	28.86	5.99	Weak	Straight and strong
Rear legs - side view	27.66	4.80	Straight	Sickle-Hocked
Rear legs - rear view	29.09	5.54	Cow-hocked	Correct
Foot angle	26.17	4.89	Narrow	Wide
Pastern	27.02	5.11	Weak	Straight and strong
Fore udder strength	27.68	5.54	Loose	Tight
Fore udder length	26.73	5.35	Short	Long
Rear udder height	26.83	5.41	Short	Tall
Rear udder width	27.14	6.03	Narrow	Broad
Suspensory ligament	28.43	5.19	Weak	Strong
Udder depth	30.47	5.44	Deep	Shallow
Udder symmetry	24.18	2.75	Not levelled front	Not leveled rear
Teats position - rear view	23.50	3.58	Far	Close
Teats Position - side view	26.74	3.92	Far	Close
Teats length	26.18	5.07	Short	Long
Front muscularity	28.61	6.13	Scarce	Developed
Rear muscularity	27.60	5.61	Scarce	Developed
Head typicality	26.44	6.21	Poor	Very good
Performance test traits:				
Average daily gain (kg/d)	1.15	0.11	0.74	1.50
SEUROP score (points)	103.3	4.09	90.0	120.0
Carcass yield (%)	56.15	1.23	51.0	60.0

**Table 2 animals-11-01340-t002:** Variance explained (Var) and percentage of the total variance explained (Var %) by the factors after rotations.

Lattent Factor	Var	Var %
FA1	2.68	0.13%
FA2	2.38	0.11%
FA3	2.03	0.10%
FA4	2.03	0.10%
FA5	1.91	0.09%
FA6	1.20	0.06%
FA7	1.25	0.06%

**Table 3 animals-11-01340-t003:** Estimates of variance components and heritability (h^2^) of analyzed traits as the means and HPD of the marginal posterior densities. Note that not all traits possess all three components (e.g., udder volume factor does not have a σ^2^_pe_, i.e., permanent environment component).

	Variance Component		Hereditability			
Traits	σ^2^_a_ ^1^	σ^2^_pe_ ^1^	σ^2^_e_ ^1^	h^2^	HPD 5 ^2^	HPD 95 ^3^
Milk traits:						
Milk yield	2.211	3.112	4.837	0.219	0.181	0.301
Fat yield	2.600 ^4^	3.190 ^4^	8.735 ^4^	0.178	0.117	0.215
Protein yield	1.895 ^4^	3.230 ^4^	10.03 ^4^	0.125	0.112	0.201
Somatic cell score (SCS, points)	0.379	0.847	1.608	0.133	0.119	0.148
Morphological aspects traits:						
Udder volume factor (F2-UV)	0.244		0.594	0.309	0.254	0.364
Udder conformation factor (F3-UC)	0.300		0.597	0.325	0.274	0.388
Rear legs factor (F7-RL)	0.208		0.661	0.238	0.181	0.241
Head typicality (HT)	13.001		21.600	0.374	0.304	0.417
Beef traits:						
Rear muscularity (RM)	9.144		18.214	0.328	0.279	0.385
Average daily gain (ADG, kg/d)	2.631		6.590	0.282	0.094	0.494
SEUROP (points)	0.529		0.863	0.376	0.184	0.567
Carcass yield (CY, %)	9.180		8.972	0.501	0.310	0.697

^1^ σ^2^_a_ is the additive genetic variance; σ^2^_pe_ is the permanent environmental variance, σ^2^_e_ is the residual variance; h^2^ is the heritability. ^2^ HPD5 is the highest posterior density region at 5%. ^3^ HPD95 is the highest posterior density region at 95%. ^4^ Variances have been multiplied by 10^3^.

**Table 4 animals-11-01340-t004:** Genetic (above the diagonal) and phenotypic (below the diagonal) correlations among milk traits, SCS, morphological, and beef traits analyzed. (within brackets). Traits that do not include zero in their HPD are reported in **bold**. Full table with HPD is reported in the Appendix A.

TRAITS ^1^	MY	FY	PY	SCS	F2−UV	F3−UC	F7−RL	HT	RM	ADG	SEUROP	CY
MY		**0.758**	**0.845**	0.069	**0.330**	**−0.444**	0.060	−0.091	**−0.458**	**−0.071**	**−0.240**	**−0.156**
FY	**0.768**		**0.824**	0.067	**0.286**	**−0.326**	0.045	**−0.136**	**−0.413**	−0.092	0.029	−0.103
PY	**0.905**	**0.766**		**0.088**	**0.289**	**−0.423**	0.099	−0.163	**−0.397**	−0.066	0.175	−0.156
SCS	**−0.149**	**−0.08**	**−0.111**		**0.246**	**0.149**	**0.190**	−0.109	**−0.156**	−0.184	−0.008	−0.259
F2−UV	**0.240**	0.172	0.211	−0.001		**−0.208**	**0.097**	0.129	−0.319	−0.121	**−0.351**	−0.359
F3−UC	−0.122	−0.067	−0.104	0.0122	0.003		0.098	0.079	**0.346**	−0.128	0.067	0.061
F7−RL	**0.02**	0.014	0.021	0.012	0.033	0.01		0.075	−0.324	0.148	−0.156	−0.159
HT	**−0.02**	−0.012	−0.023	−0.022	0.07	0.021	0.085		0.075	−0.189	0.208	0.171
RM	**−0.134**	−0.079	**−0.086**	0.001	**−0.120**	0.128	**−0.177**	**0.085**		**0.656**	**0.798**	**0.849**
ADG	−0.014	−0.108	−0.015	−0.046	−0.034	−0.037	0.043	−0.604	**0.182**		**0.839**	**0.545**
SEUROP	**−0.057**	0.006	0.034	−0.002	**−0.133**	0.025	−0.052	0.087	**0.276**	**0.621**		**0.928**
CY	−0.047	−0.024	−0.041	−0.07	−0.109	0.019	−0.041	0.06	**0.241**	**0.545**	**0.825**	

^1^ MY = Milk yield; FY = Fat yield; PY = Protein yield; SCS = Somatic cell score; F2−UV = Udder volume factor; F3−UC = Udder conformation factor; RL−F7 = Rear legs factor; HT = Head typicality; RM = Rear muscularity; ADG = Average daily gain; SEUROP = in vivo SEUROP score; CY = in vivo carcass yield.

**Table 5 animals-11-01340-t005:** Economic weights of traits as applied before the restriction for the genetic progress of target traits ^1^. The sum to 1 of the economic weights of traits considers the absolute values of the weights.

Scenario	MY	FY	PY	SCS	F2−UV	F3−UC	F7−RL	HT	RM	ADG	SEUROP	CY	Milk ^2^	Morph. ^3^	Beef ^4^
S1	0	0.24	0.46	0	0	0	0	0	0.1	0.2	0	0	0.7	0	0.3
S2	0	0.24	0.46	0 ^5^	0	0	0	0	0.1	0.2	0	0	0.7	0	0.3
S3	0	0.24	0.46	0 ^5^	0	0	0	0	0 ^5^	0.3	0	0	0.7	0	0.3
S4	0	0.217	0.433	0	0	0.07	0	0.03	0.15	0	0.05	0.05	0.65	0.1	0.25
S5	0	0.217	0.433	0	0	0.035	0	0.015	0.2	0	0.05	0.05	0.65	0.05	0.3
S6	0	0.18	0.37	0	0	0.07	0	0.03	0.15	0	0.1	0.1	0.55	0.1	0.35
S7	0	0.18	0.37	0	0	0.035	0	0.015	0.2	0	0.1	0.1	0.55	0.05	0.4
S9	0	0.24	0.46	0 ^5^	0	0 ^5^	0 ^5^	0 ^5^	0.2	0	0.05	0.05	0.7	0	0.3

^1^ Traits: MY = Milk yield; FY = Fat yield; PY = Protein yield; SCS = Somatic cell score; F2−UV = Udder volume factor; F3−UC = Udder conformation factor; RL− F7 = Rear legs factor; HT = Head typicality; RM = Rear muscularity; ADG = Average daily gain; SEUROP = in vivo SEUROP score; CY = in vivo carcass yield. ^2^ Milk traits: MY, FY, PY, SCS; ^3^ Morphological traits: F2−UV, F3−UC, F7−RL, HT; ^4^ Beef traits: RM, ADG, SEUROP, CY; ^5^ Restriction applied to target traits.

**Table 6 animals-11-01340-t006:** Economic weights of traits as applied after the restriction for the genetic progress of target traits ^1^. The sum to 1 of the economic weights of traits considers the absolute values of the weights.

Scenario	MY	FY	PY	SCS	F2−UV	F3−UC	F7−RL	HT	RM	ADG	SEUROP	CY	Milk ^2^	Morph. ^3^	Beef ^4^
S1	0	0.24	0.46	0	0	0	0	0	0.1	0.2	0	0	0.7	0	0.3
S2	0	0.186	0.356	−0.225 ^5^	0	0	0	0	0.077	0.155	0	0	0.768	0	0.232
S3	0	0.175	0.335	−0.220 ^5^	0	0	0	0	0.052 ^5^	0.218	0	0	0.730	0	0.270
S4	0	0.217	0.433	0	0	0.07	0	0.03	0.15	0	0.05	0.05	0.65	0.1	0.25
S5	0	0.217	0.433	0	0	0.035	0	0.015	0.2	0	0.05	0.05	0.65	0.05	0.3
S6	0	0.18	0.37	0	0	0.07	0	0.03	0.15	0	0.1	0.1	0.55	0.1	0.35
S7	0	0.18	0.37	0	0	0.035	0	0.015	0.2	0	0.1	0.1	0.55	0.05	0.4
S8	0	0.153	0.294	−0.134 ^5^	0	0.160 ^5^	0.052 ^5^	0.015 ^5^	0.128	0	0.032	0.032	0.581	0.227	0.191

^1^ Traits: MY = Milk yield; FY = Fat yield; PY = Protein yield; SCS = Somatic cell score; F2−UV = Udder volume factor; F3−UC = Udder conformation factor; RL− F7 = Rear legs factor; HT = Head typicality; RM = Rear muscularity; ADG = Average daily gain; SEUROP = in vivo SEUROP score; CY = in vivo carcass yield. ^2^ Milk traits: MY, FY, PY, SCS; ^3^ Morphological traits: F2−UV, F3−UC, F7−RL, HT; ^4^ Beef traits: RM, ADG, SEUROP, CY; ^5^ Restriction applied to target traits.

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
