# Peer review of "Selection Response Due to Different Combination of Antagonistic Milk, Beef, and Morphological Traits in the Alpine Grey Cattle Breed"

_animals, 2021, doi:10.3390/ani11051340_

Round 1
Reviewer 1 Report
General comments
The manuscript deals with the development of an aggregated genetic index in a local dual-purpose breed, by testing some scenarios.
Although not very original, the idea presented is nevertheless an important contribution in the field.
Any attempt to improve the management of local breeds is appreciable because biodiversity preservation becomes a sustainable process only if it is done by balancing productive, functional and environmental adaptation traits.
In this regard, the manuscript acquires a significance that goes beyond the apparent local interest of the work.
In general, the paper is well structured, very well written, with extreme clarity and accuracy.
The introduction defines the problem adequately, the goal of the work and the experimental design are consistent.
Also the materials and methods section is well structured, clear and complete.
The used statistical approach for the analysis is appropriate and the amount of data sufficient to obtain robust responses.
Results section provides an adequate presentation of the research results, presented clearly and completely, and discussed comparing them with the available literature in a correct way.
The conclusions are fully supported by the obtained results.
The manuscript only needs minor corrections; I have only a few comments and suggestions reported below.
Specific comments:
L 93-94: specify the interval between test days
L 97: delete "and" after ADG and replace with a comma
L 72: To complete the breed description, I believe that a brief description of the breeding system adopted for this breed is useful.
L 111: the choice to include lactations with the first control within 45 days of lactation should be justified. I do not know the shape of the lactation curve of this breed, but in most dairy cows at 45 DIM the lactation peak has already occurred. The use of lactations with the first test day so far from calving could introduce an important bias into the model, particularly when using polynomials to model milk traits. If valid reasons exist, they should be considered and at least briefly discussed in the paper.
L 122 - 127: considering the generalist nature of the journal, I think it is better to write briefly, why this model of analysis is adopted and how the model works. Furthermore, the reason why only some of the extracted factors are chosen for the subsequent analysis should be explained and discussed in detail.
L 128 - 138: This part should be moved to the results section.
L 183 - 184: it seems to me that this concept has not yet been expressed.
L 202: am instead of am.
L 306 – 307: variance explained by the factors after rotation should be tabulated.
L 321: I think that, for a better understanding (and evaluation) of the multivariate structure of the analyzed traits, figure 1 should be replaced by a table showing all the estimated loadings by the model, the eigenvalues and the percentage of variance explained by each extracted factor. In order to highlight the factorial structures, loadings >|0.45| can be reported in bold.
L 337: the confidence limits in the table for Fat and Protein yield seem to be wrong.
L 463 e 467: In this section, the term factor score is used as a synonym of loading in the description of the extracted factors, this is not correct, the loading always refers to the variables, while the scores refer to the samples.
Author Response
We thank the reviewer for their appreciative words and their thorough evaluations.
Specific comments:
L 93-94: specify the interval between test days
Thanks for the correction, interval between test days is 4-5 weeks and was added as requested (please see lines 111).
L 97: delete "and" after ADG and replace with a comma
We thank the reviewer for the correction, which was implemented
L 72: To complete the breed description, I believe that a brief description of the breeding system adopted for this breed is useful.
We agree with the observations, a brief description was added (line83-86)
L 111: the choice to include lactations with the first control within 45 days of lactation should be justified. I do not know the shape of the lactation curve of this breed, but in most dairy cows at 45 DIM the lactation peak has already occurred. The use of lactations with the first test day so far from calving could introduce an important bias into the model, particularly when using polynomials to model milk traits. If valid reasons exist, they should be considered and at least briefly discussed in the paper.
We added a brief explanation to support our reasoning (lines 132 133). Thanks for the observations
L 122 - 127: considering the generalist nature of the journal, I think it is better to write briefly, why this model of analysis is adopted and how the model works. Furthermore, the reason why only some of the extracted factors are chosen for the subsequent analysis should be explained and discussed in detail.
According to the reviewer's comment, a sentence was added in order to explain better why the authors used the varimax procedure. (150-156)
L 128 - 138: This part should be moved to the results section.
We added this paragraph to the “results” section according to the reviewer comment However, in order to simplify the readability of the manuscript, we preferred to keep this result also in this section (and in the “models” section”), as part of the subsequent analyses would be harder to follow if it was not reported here.
L 183 - 184: it seems to me that this concept has not yet been expressed.
The sentence was rewritten, please see lines 216 – 217.
L 202: am instead of am.
We thank the reviewer for pointing out the typo, which was corrected.
L 306 – 307: variance explained by the factors after rotation should be tabulated.
According to the reviewer's comment, Table 2 was added.
L 321: I think that, for a better understanding (and evaluation) of the multivariate structure of the analyzed traits, figure 1 should be replaced by a table showing all the estimated loadings by the model, the eigenvalues, and the percentage of variance explained by each extracted factor. In order to highlight the factorial structures, loadings >|0.45| can be reported in bold.
We added the requested figure to increase the readability of the paper.
L 337: the confidence limits in the table for Fat and Protein yield seem to be wrong.
We thank the reviewer for finding this typo, which we corrected. We revised the figure according to reviewers 1 and 2: table was redone, splitting the residual component into its genetic, permanent and residual components.
L 463 e 467: In this section, the term factor score is used as a synonym of loading in the description of the extracted factors, this is not correct, the loading always refers to the variables, while the scores refer to the samples.
We thank you for the correction, the mistake was resolved by changing the wording throughout.
Reviewer 2 Report
This manuscript aims at investigating genetic background, in particular the genetic correlations among antagonistic traits, in a local cattle breed in order to optimize the weights of an aggregate selection index. The paper is rather interesting, maybe slightly long, but rigorous and detailed. Below I have listed few minor comments, remark/suggestions
Lines 114-115: Something about pedigree completeness: 6 generation were traced back. Were those six ones complete generations?
Lines 140-141: rear muscularity and head typicality are two linear scores? It is interesting this aspect of HT but what is actually assessed (in the results I saw that did not present relevant genetic correlation with the other traits).
Lines 198. Have the authors considered to treat the classifiers as random effect instead? Please the authors to explain the reason to include DIM class for evaluating all morphological traits (including HT, RM). Please replace Cn with Cj at line 199
Lines 236-239. Please the authors to be more specific, it is not clear from the text
Lines 253-257. Please revise this period. Some sentences are too long with many subordinates. Moreover, put boldface the matrix notation.
Lines 261. Please define σpi; or is it actually σi ?
Genetic parameter paragraph , page 8. For milk traits, have you estimated repeatability too?
Line 371 please a full stop after zero(b).
Line 375. Replace figure 3M with figure 2
A general comment about discussion section in particular the conclusions
The authors have run different simulation, but did not offer to the readers their preference on the best scenario. The conclusion that they draw is about the current one that is not defined appropriate in the long run. According to their results this is undoubtedly true. Clearly, the economic values of a standard deviation of different traits are not equally predictable. For some traits is very difficult to hypothesize them (e.g. composiste morphological traits, or HD) rather than milk/meat incomes. However according to my opinion, after this simulation an evaluation of the best (in addition to the worst) scenario should be provided.
Table 3. I had difficult to read Table 3. Apart from the font size, the presence of HDP5-95 made it difficult to find the searched values. I would suggest to get rid of HDP interval and report in the table the estimate correlation coefficients only, marking opportunely those that include zero in their HPD interval. The authors may think to attach the full table in a more readable format as supplementary materials.
Author Response
We sincerely thank the reviewer for their work and for their kind words.
Lines 114-115: Something about pedigree completeness: 6 generation were traced back. Were those six ones complete generations?
Yes, indeed those were complete generations – the sixth back was the latest complete generation.
Lines 140-141: rear muscularity and head typicality are two linear scores? It is interesting this aspect of HT but what is actually assessed (in the results I saw that did not present relevant genetic correlation with the other traits).
We thank the reviewer for the observation, a brief description about the why these traits have been included has been written.
Lines 198. Have the authors considered to treat the classifiers as random effect instead? Please the authors to explain the reason to include DIM class for evaluating all morphological traits (including HT, RM). Please replace Cn with Cj at line 199
We understand the reviewer question: however, since only 4 classifiers are included and the number of records is homogeneous among the classifiers’ levels, we decided to include the classifiers only as fixed. For what concerns DIM, this factor greatly impacts overall body morphology condition of the animals: for HT, while it is true that the effect was only marginally significant – the reviewer is correct, we do not indeed expect DIM to affect greatly HT – the effect was retained mostly in order to balance the estimation of the residual components’ during the bitrait analysis.
Lines 236-239. Please the authors to be more specific, it is not clear from the text
Sentence was rewritten L 282
Lines 253-257. Please revise this period. Some sentences are too long with many subordinates. Moreover, put boldface the matrix notation.
Sentence was rewritten and subdivided: thanks for correction
Lines 261. Please define σpi; or is it actually σi ?
We thank the reviewer for catching this mistake, which we corrected.
Genetic parameter paragraph , page 8. For milk traits, have you estimated repeatability too?
According to reviewer 1 and 2 table was rewritten (now it is table 3) , splitting the residual component into a residual, genetic and permanent ones and correcting all typos
Line 371 please a full stop after zero(b).
We corrected the typo that the reviewer noticed.
Line 375. Replace figure 3M with figure 2
We corrected the mistake that the reviewer noticed.
A general comment about discussion section in particular the conclusions
The authors have run different simulation, but did not offer to the readers their preference on the best scenario. The conclusion that they draw is about the current one that is not defined appropriate in the long run. According to their results this is undoubtedly true. Clearly, the economic values of a standard deviation of different traits are not equally predictable. For some traits is very difficult to hypothesize them (e.g. composiste morphological traits, or HD) rather than milk/meat incomes. However according to my opinion, after this simulation an evaluation of the best (in addition to the worst) scenario should be provided.
We added a paragraph following the reviewer suggestions, Please see line .711 720
Table 3. I had difficult to read Table 3. Apart from the font size, the presence of HDP5-95 made it difficult to find the searched values. I would suggest to get rid of HDP interval and report in the table the estimate correlation coefficients only, marking opportunely those that include zero in their HPD interval. The authors may think to attach the full table in a more readable format as supplementary materials.
We totally agree with the reviewer’s conclusion: table has been partially rewritten (now is table 4); full table is now in supplementary material.
Reviewer 3 Report
Dear Authors, I read your publication with interests. Alpine Grey cattle is of great importance for grazing in Alpine pasture and producing high quality meat and milk products. The selection pressure on productive traits lead to worsening, in my opinion, mainly body conformation as well as functional traits. Over the past 50 years, many cattle breeds have lost their original, mainly dual-purpose character. By selecting for the increase of milk yield, health parameters and body structure were worsened. Metabolic diseases such as ketosis and lameness appeared, as well as high SCS. Cows went out to pasture less and less. The presented publication shows that these changes also affected the Alpine Gray breed, which is mainly used for pasture. The body conformation should enable cows of this breed to easily move on steep alpine slopes and graze also at an altitude of more than 2000 m. Too high economic weight of milk characteristics in the selection index used for this breed (S1) led to the deterioration of the meat and health characteristics (SCS). Therefore, the attempt to change this unfavorable tendency by using different economic weights for individual features in the selection index is perfectly appropriate. The presented work also introduces a detailed analysis of the relationships between the milk yield, fleshiness and body conformation. This makes it easier to follow the each scenarios for individual traits included in the selection index. It is regrettable that functional traits such as fertility and health, better related to the meatness traits and body conformation, were omitted. It would then be possible to reduce the weight of milk characteristics by a further 5-10%. In the Simmental breed, they constitute only 38%. I agree with the authors of the publication that the most appropriate scenarios would be the S6 and S7 with a reduced importance of the milk characteristics. They can also be helpful by improving the SCS. Also, in other cattle breeds, it has been noticed that the meat characteristics are better related to the functional ones. Therefore, I would suggest exploring this group of resilience related traits in the future and including them in the selection index, lowering the importance of milk performance traits. In line 31 should be "first". Well done.
Author Response
We sincerely thank the reviewer for their work and for their kind words.
We corrected the typo. We agree with the reviewer on the importance of our work and in the future, we would definitely like to extend our analysis to the suggested traits. We also added a brief paragraph about scenario 7.